## Research Article

perinatal depression; thinking healthy program; psychological intervention; adaptation; nonspecialist providers; Nepal

**Corresponding author:**
Prasansa Subba;
Email: p.subba@liverpool.ac.uk

# Thinking local, thinking healthy: Cultural and contextual adaptation of the Thinking Healthy Programme in Nepal

Prasansa Subba[1,2,3] , Pragya Shrestha[2], Rupa Rai[4,5], Shristi Subedi[2,6], Nagendra Luitel[2] , Atif Rahman[1] , Siham Sikander[1] and Najia Atif[7]

[1]Department of Primary Care and Mental Health, Faculty of Health and Life Sciences, University of Liverpool, UK; [2]Research Department, Transcultural Psychosocial Organization Nepal, Nepal; [3]Department of Public Health, Faculty of Health and Medical Sciences, University of Copenhagen, Denmark; [4]Programme Department, Transcultural Psychosocial Organization Nepal, Nepal; [5]Tribhuvan University - Padma Kanya Multiple Campus, Nepal; [6]Department of Health Science, Manmohan Memorial Purwanchal College, Nepal and [7]Human Development Research Foundation, Pakistan

## Abstract

*Background:* The thinking healthy program (THP) is an evidence-based psychological intervention for perinatal depression designed for delivery by nonspecialist health workers. To ensure its relevance in Nepal, we adapted THP using the mental health Cultural Adaptation and Contextualization for Implementation (mhCACI) framework. *Methods:* Using mhCACI's 10-step process, we applied a participatory approach involving a multidisciplinary team to adapt both content and implementation strategies. A qualitative study nested within a pilot trial was conducted to assess feasibility and acceptability of adapted THP through in-depth interviews with perinatal women (n = 20), family members (n = 11) and focus group discussions with Female Community Health Volunteers (FCHVs) (n = 16). *Results:* FCHVs were selected as delivery agents. Implementation adaptations included reducing the number of THP sessions from 16 to 8, integration of additional 2.5-day Foundational Helping Skills training and skill-based training methods. Manual revisions included simplified language, cultural idioms, visual aids and locally relevant examples. Referral pathways for gender-based violence, suicide and severe mental illness were included. The adapted THP was well received by providers and recipients. *Conclusion:* The adaptation demonstrates how global interventions can be contextually tailored for low-resource settings while preserving therapeutic integrity, offering a scalable model for community-based mental health care.

## Impact statement

This study describes the cultural and contextual adaptation of Thinking Healthy Programme (THP), a psychological intervention for perinatal depression in Nepal, using the mental health Cultural Adaptation and Contextualization for Implementation framework. It provides a practical guide for adapting psychological interventions in low-resource settings while maintaining theoretical fidelity. By integrating the intervention within Nepal's existing health system and using trusted community-based providers (Female Community Health Volunteers), the study enhances the feasibility, acceptability and sustainability of perinatal mental health care. The findings offer valuable insights into the balance between cultural fit and intervention integrity, which is an ongoing challenge in global mental health implementation. This article is relevant to researchers, implementers and policymakers seeking to implement evidence-based interventions in diverse cultural contexts.

## Introduction

Experience of depression (Accortt and Wong, 2017; Lim, 2021) and psychiatric admission (Howard and Khalifeh, 2020) among women is significantly higher during pregnancy and postnatal period than other times. Perinatal depression refers to prolonged persistence (more than 2 weeks) of symptoms of irritability, low mood, extreme worry affecting the woman's functioning and wellbeing during pregnancy and 1 year after childbirth (Kroh and Lim, 2021). Estimated 26% of women globally (Al-Abri et al., 2023) and 28% in South Asia experience perinatal depression (Thakuri et al., 2025).

Treatment options for perinatal depression in primary healthcare settings can be broadly categorized into pharmacological and psychological approaches. Pharmacological treatment is recommended for moderate to severe depression, or when psychotherapy is not effective (Kittel-Schneider et al., 2022). The World Health Organization recommends psychological interventions

as a first line of treatment for perinatal depression (WHO, 2015). While there are many psychological intervention approaches, cognitive behavioral therapy (CBT) is the most widely used approach to treat perinatal depression (Subba et al., 2025). The CBT approach focuses on identifying and transforming unhealthy thoughts into healthier ones, leading to improved emotions and behaviors (Kroh and Lim, 2021).

One such intervention that is built on the theoretical stance of CBT is the THP (WHO, 2024). Multiple randomized controlled trials on THP have demonstrated its effectiveness reducing depressive symptoms and disability while also improving their functioning and social support (Rahman et al., 2008; Sikander et al., 2019; Aguilar et al., 2023; Botyar et al., 2024). Importantly, THP has shown to be effective when delivered by nonspecialist providers with no prior mental health training across a range of global settings (Atif et al., 2017; Fuhr et al., 2019; Sikander et al., 2019; Boran et al., 2023a; Botyar et al., 2024; Ng'oma et al., 2024).

There is no national data for prevalence of perinatal depression in Nepal, but studies conducted in different parts of the country have somewhat estimated to be between 24 and 41% (Dielemans et al., 2024; Wasti et al., 2024; Thakuri et al., 2025). The current antenatal and postnatal services available at the health facilities fail to address mental health issues and are exclusively focused on physical health. While there may be many other factors, the most prominent include inadequate financial allocation and a lack of human resources for mental health (Rai et al., 2021). In such a case, the THP holds promise as a scalable and cost-effective solution (Rahman et al., 2008; Fuhr et al., 2019).

However, an intervention developed in one context might not be relevant to other cultural settings (Sit et al., 2023). Conversely, culturally adapted interventions are more likely to be acceptable, feasible and effective (Castro et al., 2004; Harper Shehadeh et al., 2016). In Nepal, where social and cultural norms deeply shape perceptions of mental illness and help-seeking behaviors, ignoring these nuances may limit uptake and impact (Kaphle et al., 2013; Subba et al., 2024).

Recognizing the importance of cultural and contextual relevance, we employed the mental health Cultural Adaptation and Contextualization for Implementation (mhCACI) framework developed by Sangraula et al. (2021) to adapt the THP. The mhCACI framework consists of 10-step adaptation process and emphasizes not only content adaptation but also the alignment of implementation strategies with local systems, structures, and values. This article details the cultural and implementation adaptation of the THP for perinatal depression in Nepal, highlighting both the process and rationale for modification.

## Methods

### Setting

This study was conducted in Jhapa, a district in Koshi Province, one of the seven provinces of Nepal. As part of the adaptation process, an expert consultation workshop was held in Kathmandu, the capital city of Nepal. Nepal is a low-income country with a history of decade long armed conflict between 1996 and 2006, natural disasters (notably two devastating earthquakes in 2015) and ongoing political unrest. Despite the increasing mental health problems, it is estimated that there are only 250 psychiatrists, 37 psychologists, 75 psychiatric nurses, approximately 1,300 para professional counselors serving a population of 30,000,000 and most of the specialists are centered in big cities (Sharma et al.,

2024). Despite mental health services being listed in the universal health coverage (Nepal Law Commission, 2018) not all health facilities are equipped with trained human resources or mental health services. The Transcultural Psychosocial Organization (TPO) Nepal is one of the leading and experienced nongovernmental organizations working in mental health in conflict affected, post disaster settings in Nepal since 2005. TPO Nepal also has a strong track record of adapting evidence-based psychological interventions such as the Problem Management Plus (Sangraula et al., 2021) and Interpersonal Therapy (Clarke et al., 2020). The adaptation of THP intervention was also led by the team at TPO Nepal.

### Thinking Healthy Programme

The THP is a fully manualized CBT-based intervention that is a part of the WHO's mental health gap action program (WHO, 2024). The original version comprises of 16 sessions in total – an opening session followed by 5 modules each consisting of three sessions on woman's personal health, her relationship with the baby and her relationship with the family. The content focus on vicious cycle of negative thoughts, its impact on feeling and behavior that are complemented with relevant illustrations. The reference manual is designed for the nonspecialist providers and lists step-by-step instructions to deliver THP sessions. The "health calendar" is a workbook given to the THP recipients. It contains sessions' key messages and health charts with goals for helpful activities, set in collaboration with the recipients. The programme has been rigorously evaluated in Pakistan and implemented in several countries (Fuhr et al., 2019; Sikander et al., 2019; Rahman et al., 2021; Boran et al., 2023b).

### Adaptation framework

The adaptation process was guided by the mhCACI framework developed by researchers at TPO Nepal. The mhCACI incorporates a systematic, iterative 10-step process divided into three phases that involves identifying the mechanisms of action of the intervention, assessing local understandings of distress and treatment and modifying both content and delivery strategies accordingly. The framework is grounded in the ecological validity model (EVM) (Bernal et al., 1995). Utilizing the eight key dimensions of EVM such as language, persons, metaphors, content, concepts, goals, methods and context, the framework serves as a guide for tailoring interventions to fit within the cultural and contextual factors of the target population. The framework prioritizes participatory engagement with community stakeholders and encourages rigorous documentation at each step to ensure transparency and replicability. By prioritizing both fidelity to core intervention mechanisms and cultural/contextual fit, mhCACI enhances the acceptability, feasibility and potential scalability of mental health interventions in resource-limited settings (Sangraula et al., 2021).

### Adaptation phase 1: Preconditions

#### Step 1: Identifying mechanisms of action
The mhCACI defines key mechanisms of action as the core therapeutic components or techniques through which distress is addressed and wellbeing is promoted. Preserving these mechanisms is essential to maintaining the intervention's integrity. Throughout the adaptation process, the team based in Nepal frequently scheduled a call with the development team (co-authors Najia Atif [NA],

**Table 1.** Mechanism of action of the THP intervention (Rahman et al., 2025)

| Mechanisms | Description of mechanism | Implementation of mechanism |
|---|---|---|
| Empathetic listening | THP facilitators develop foundational counseling skills to better understand and identify the woman's concerns. | Sessions 1 through 8 |
| Social support | THP facilitators involve family members in sessions to emphasize the importance of their support and identify practical ways to provide it. | Sessions 4 and 7 |
| Psychoeducation | Participants learn about the connection between thoughts, feelings and behaviors, and how unhelpful thinking styles can impact their wellbeing. | Sessions 1 through 8 |
| Thoughts restructuring | Participants explore their vicious cycle of depression. They learn to break this cycle by identifying and replacing unhealthy thoughts with healthier alternatives. | Sessions 1 through 8 |
| Behavior activation | Participants set collaborative health-related goals. They track their activities and moods to build awareness and promote engagement in positive behaviors. | Sessions 1 through 8 |
| Problem solving | Participants identify specific problems, explore their impacts and develop strategies to manage or resolve them. They are encouraged to implement solutions discussed during the session. | Sessions 1 through 8 |

Siham Sikander [SS] Atif Rahman [AR]) to identify and retain these mechanisms of action. See Table 1

### Step 2: In-depth literature review

A systematic review of existing literature of psychosocial interventions for perinatal depression, and delivery agents was conducted as a part of the adaptation process (Subba et al., 2025). Further, a review of scholarly articles and Nepal government's guideline on maternal and child health (MCH), mental health training documents, along with community health strategy for maternal health was conducted to understand the maternal health and maternal mental health landscape in the Nepali context.

### Adaptation phase 2: Preimplementation

### Step 3: Training of trainers

A cascaded training and supervision model was employed for the THP (Zafar et al., 2016; Atif et al., 2019). A mental health specialist, a master trainer of THP and coauthor (NA) conducted a 4 day (30 h) virtual Training of Trainers (ToT) for authors Prasansa Subba (PS1) and Pragya Shrestha (PS2), who subsequently led the adaptation process. The training was given on the original THP manual and was conducted in English. Both PS1 and PS2 are experienced mental health researchers with master's degrees in public mental health and psychology. The ToT included lectures, discussions and role-plays, covering an overview of the THP and perinatal depression on Day 1; CBT principles and THP

prerequisites on Day 2; session content and delivery on Day 3; and implementation strategies with practice sessions on Day 4.

### Step 4: Translation of the manual

A bilingual translator, trained in psychosocial counseling and experience in mental health research, translated the English version of WHO THP manual into Nepali. Her background provided the necessary expertise to interpret and contextualize the psychological concepts appropriately within the local idiomatic and cultural framework. The translated material underwent a thorough proofreading process conducted by a Nepali language expert. Visual elements of the manual were illustrated by a local artist with more than 5 years of experience working in the mental health field.

### Step 5: Expert read through

Authors PS1 and PS2 regularly reviewed the translated manual to ensure the simplification of language and the content to make it more suited for Female Community Health Volunteers (FCHV) with limited education. Two psychiatrists critically reviewed both English and Nepali manuals to ensure whether the content reflected the Nepali context.

### Step 6: Formative qualitative study

Key informant interviews (KII) were conducted with FCHVs (n = 2) and health workers (n = 6) in Kathmandu followed by four Focus Group Discussion (FGD) with FCHVs (n = 41) in Jhapa district. Separate topic guides were developed for health workers, and FCHV. PS1 and PS2 conducted KIIs and FGDs. KIIs lasted 1–1.5 h, and FGDs lasted 2–2.5 h. All qualitative data were collected in Nepali, audio-recorded and transcribed. We used a thematic analysis approach to analyze data from the formative study. Key themes include – FCHVs' responsibilities, local maternal health practices, community perceptions of maternal depression, and stigma.

### Phase 3: Implementation

### Step 7: Practice rounds

PS1 and PS2 trained in Step 2 of the adaptation process practiced at least two modules of the THP with four perinatal women. Module 1 was practiced with two pregnant women and Module 2 was practiced with two postnatal women. The practice round lasted for 3 months. Both local trainers received regular supervision from NA. The practice sessions were conducted to pilot the translated manual and to gather feedback on comprehensibility and applicability of the training content. Changes required to the manual or implementation strategy were reflected after each practice session.

### Step 8: Adaptation workshop

A one-day adaptation workshop was held with a multidisciplinary team comprising of experts in mental health research (n = 4), maternal health (n = 4), psychiatry (n = 2), psychology (n = 2), psychosocial (n = 2) and medical anthropology (n = 2). Prior to the workshop, the manual was divided into three thematic areas focusing on – (a) women's personal health, (b) their relationship with the baby and (c) their relationship with the family. Each invited participant was assigned one section of the draft adapted THP manual for review. During the workshop, feedback was collected on manual content, and participants discussed locally relevant examples of negative thoughts, emotions and behaviors related to the perinatal period, along with culturally appropriate alternatives, common practices and anticipated implementation challenges.

## Step 9: Implementation, supervision and process evaluation

The adapted THP was evaluated through a pilot cluster randomized controlled trial (cRCT) designed to assess its feasibility and acceptability, as well as the suitability of FCHVs as delivery agents. The trial also included an assessment of FCHVs' competency, which is reported elsewhere (manuscript forthcoming). Given the established role in Nepal's health system as champions of community MCH services, FCHVs were selected as delivery agents for THP. In the trial, the intervention arm received THP sessions delivered by trained FCHVs, while the control arm received enhanced usual care, which involved screening and referral of pregnant women with depressive symptoms to primary healthcare facilities offering basic mental health services. FCHVs underwent 2.5 days training on Foundation Helping Skills (FHS) (15 h) (Watts et al., 2021; Pedersen et al., 2023) and 5 days on THP (30 h).

Pregnant women scoring ≥10 on the validated Patient Health Questionnaire (Kohrt et al., 2016) were enrolled and received home-based THP sessions from trained FCHVs. Of the 17 FCHVs trained, only 14 FCHVs delivered THP. Mandatory monthly group supervision, and individual in-person or phone-based supervision as per the need were provided to ensure quality of care. Supervision notes were maintained throughout implementation process to inform potential adaptations to the intervention manual.

A qualitative study nested within the pilot cRCT was conducted at the end of the intervention implementation to assess the feasibility and acceptability of the THP. In-depth interviews (IDIs) were held with 20 women (17 completed all sessions) and 11 family members (seven attended at least one session). Two FGDs were also conducted with six FCHVs each. Interviews were conducted at participants' homes and FGDs at the health facility. IDIs and FGDs were facilitated by authors PS1 and Shristi Subedi (SS), lasted approximately 1 and 2 h, were conducted in Nepali, audio-recorded and transcribed. We followed the five steps of framework approach where we first (a) familiarized with transcripts, (b) developed a thematic framework, (c) indexed data according to emerging themes, (d) charted data within the framework and finally and (e) mapped and interpreted relationships between themes in relation to the study objectives (Ritchie and Spencer, 2002).

## Step 10: Recustomization of intervention

Final modifications to the THP manual were informed by supervision notes and findings from the qualitative study post THP implementation. These refinements ensured that the adapted intervention was responsive to real-world challenges and aligned with feedback from both implementers and recipients.

## Results

Key adaptations made to the intervention have been organized under three key phases as outlined in the mhCACI framework. See Table 2 for a summary of key adaptations and supplementary file for details of adaptation record.

### Phase I: Preconditions

To align with the FCHVs' community, visit schedules and responsibilities, THP sessions were restructured from five to two modules, delivered over four antenatal and four postnatal visits. Visits 1–7

**Table 2.** Summary of key adaptations according to the mhCACI framework

| Domains | Operationalization | Phase 1 (desk review) | Phase 2 (preimplementation) | Phase 3 (implementation) |
|---|---|---|---|---|
| Language | Expression, gestures, verbal style, local idioms, | Culturally acceptable idioms of distress, such as *"maan ko samasya"* ("heart-mind problems") and "tension" were used to express depressive symptoms. | Manual translation to Nepali by a professional translator. | "Healthy and unhealthy thoughts" replaced by "positive and negative thoughts" |
| | | | Simplification of terms using description for terms such as 'psychoeducation', 'depressive disorder', 'cognitive behavior therapy', 'mhGAP Intervention Guide', 'evidence-based psychosocial intervention', 'empathy'. | |
| Metaphors | Symbols, sayings/ proverbs | X | Culturally relevant illustrations, including images of women in traditional attire, were added to enhance relatability. | X |
| | | | The CBT cycle was also presented pictorially for better understanding. | |
| | | | Local proverbs – such as *"nani janmanu kaile kaile kokro buncha ahile"* (akin to "Don't count your chickens before they hatch") – were used to reflect common beliefs that discourage women from preparing for childbirth in advance. | |
| Content | Incorporation of local data, relevant content, practices | Additional information and statistics on perinatal depression in Nepal added in the background section. | Additional 2.5 training on communication based on the Foundation Helping skills given as a prerequisite for THP. | Benefits of activities in the health calendar added to the manual. |
| | | | Culturally relevant examples of negative thoughts and cognitive restructuration added. Examples of cultural preference to son added. | A closing session was added to help women reflect on progress, anticipate future challenges and formally conclude their engagement with the FCHV. |

*(Continued)*

**Table 2.** (*Continued*)

| Domains | Operationalization | Phase 1 (desk review) | Phase 2 (preimplementation) | Phase 3 (implementation) |
|---|---|---|---|---|
| Concepts | Incorporation of culturally relevant resources | Diet chart with examples of locally available food and local eating habits were added from the government FCHVs' manual | Given the pivotal role of mother-in-law in perinatal care in Nepal, her engagement in the THP session was highlighted. | Information on referral centers for gender-based violence, suicide, family planning, locally available financial and medical support services added. |
| | | | Schedule of postnatal sessions adjusted to the cultural practice of relocation to maternal home for postnatal care. | |
| People | Delivery agent | Identification of Female Community Health Volunteer (FCHV) for THP delivery | Manual structure was adapted. For example, session overviews were added; long paragraphs were converted into bullet points. | X |
| Methods | Implementation strategies | Restructuration of THP from 16 to 8 sessions to align with the government's protocol for FCHVs. | Interactive participatory training methods applied instead of didactic methods. | Mandatory practice sessions for both modules added before THP delivery. |
| Context | Incorporation of contextual factors to boost understanding | An additional section explored how stigma impacts help-seeking, treatment adherence and community participation. | Contextual risk factors (lack of family support, early marriage, alcohol abuse, absent husband due to foreign employment, heavy household responsibilities) were incorporated | Notes were added prompting FCHVs to ask women what physical activities would be feasible for them, especially when standard exercises felt inappropriate or impractical. |
| | | The overall aim of THP was also reframed to emphasize maternal well-being, rather than directly naming perinatal mental health, to reduce stigma and increase acceptability. | Additional subsection on stigma and how its impact on delaying help-seeking behavior, treatment adherence and participation in community activities. | |
| | | | Role of family members and practical examples of how they can support (such as allowing time to rest, feeding nutritious food, accompanying to the health facility, helping in daily chores) were added | |
| Goals | Reflecting knowledge of culture and values | X | A section was included to normalize pregnancy as a natural experience and counter the cultural narrative of shame or silence surrounding it, encouraging open discussion | Encouraged women to resume meaningful daily activities stopped during pregnancy or postpartum. |

followed the original content, while visit 8 focused on reviewing key learnings and planning for future challenges. Postnatal visits began 2 weeks after delivery and were held weekly, accommodating the cultural norm of relocating to maternal homes for postnatal care around the third month. Additional adaptations included incorporating local prevalence data, replacing clinical terms with culturally resonant idioms like *"maan ko samasya"* ("heart-mind problems") (Kohrt and Harper, 2008) and *"tension"* (Clarke et al., 2014; Chase, 2021; Jordans et al., 2021) and reframing the intervention's aim to promote overall maternal wellbeing rather than addressing perinatal mental health problems.

### *Phase II: Preimplementation*

Key adaptations in the preimplementation step included translation, language simplification and adaptation of the THP content to match the FCHVs' competency. In this stage, instead of healthy thought, positive thought was used throughout the manual and "visits" was used instead of sessions. Similarly, rather than using complex technical jargon, description was used. For example, "perinatal depression" was explained as prolonged sadness, disinterest and stress and "cognitive behavioral therapy" was framed as a method to shift unhelpful thoughts and behaviors to foster positive emotions. Further, a pictorial presentation using clear expressions were used to elucidate the cycle of thoughts–emotions–behavior. See Figure 1.

Formative research identified the need for additional training in communication and counseling skills to prepare FCHVs to deliver psychosocial support effectively. Therefore, a 2.5-day supplemental

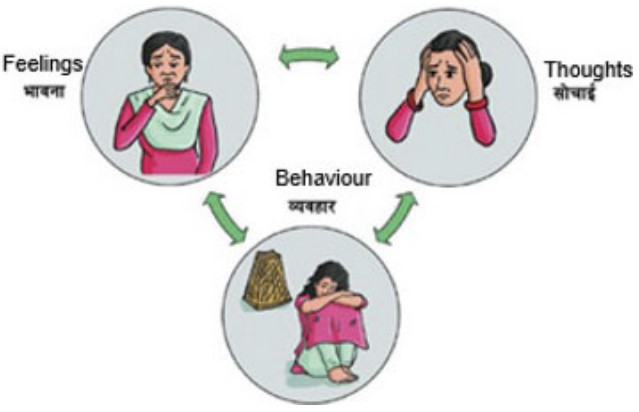

**Figure 1.** Illustration of the relationship of thoughts–emotion–behavior cycle.

**Table 3.** Key findings of qualitative study

| Themes | Key findings | Supporting quotes |
|---|---|---|
| Trusted Care Through Established Community Roles | • FCHVs were trusted and accepted due to their established community role.<br>• Familiarity between FCHVs and participants enhanced comfort and ease.<br>• FCHVs' visits for THP delivery were perceived as part of routine health interactions.<br>• Delivery of psychosocial support felt natural and nonstigmatizing. | *"I was happy when sister (FCHV) came to visit me. It was easy to talk to her. I didn't feel uncomfortable."* – Perinatal woman, 33 years<br>*"She (FCHV) used to come from time to time. Such services were not available in the past. I felt that they (FCHV) care about pregnant women when they came to give advice in our household."* – Family member (husband), 29 years<br>*"I didn't know where to go, or whom to talk to. She told me that I can call or meet her anytime. After that I felt everything will be okay."* – Perinatal woman, 33 years |
| Culturally relevant and clear content | • Content and examples were perceived as culturally and contextually relevant.<br>• Physical exercises were highly valued for maternal well-being.<br>• Visual aids (pictures) improved comprehension and engagement.<br>• FCHVs suggested simplifying formal language (e.g., "positive/negative" to "good/bad"). | *"I used to have a racing heartbeat. At that time, she (FCHV) taught me to take a deep breath in and deep breath out. This helped me to feel relaxed."* – Perinatal woman, 26 years<br>*"I thought to myself that the woman (referring to the picture in the manual) might have had similar tension as mine that's why she looked sad."* – Perinatal woman, 36 years<br>*"We are FCHVs from a village so if you are targeting us, the language should be simple."* – FCHV, 53 years |
| CBT skills in action | • FCHVs gained confidence in applying CBT techniques with practice.<br>• Identifying causes of negative thoughts helped FCHVs in reframing them positively.<br>• Women valued learning about the impact of thoughts on their and their baby's health.<br>• Perinatal women appreciated the education on how their negative thoughts could impact their health and their baby's development. | *"At first, it (the CBT concept) was difficult for us to understand. Once we understood how negative thoughts affect us and were able to identify them then we understood the process of converting into positive. Only when we were clear about the concept, were we able to explain. Afterward, it was not that difficult but initially, it took us time to understand.* – FCHV, 55 years<br>*"It helped me change my thought processes. If it had not happened, it would be better not to ruminate about it. There's no use thinking about it (negative things). It will only affect me. That's what I learned."* – Perinatal woman, 30 years |
| Session adequacy | • Eight sessions were seen as sufficient by most, although some wanted more antenatal sessions as antenatal phase was seen as the most critical period for support. | *"For me it was enough because I learned that I should not think negative. I started thinking positively. Hence, for me the number of sessions were adequate."* – Perinatal woman, 26 years |

training on FHS was added. Similarly, additional content on stigma and risk factors that emerged from the formative study were also added in the background section.

## Phase III: Implementation

Practice rounds allowed local trainers to gain hands-on experience and identify implementation challenges. For example, one participant's disclosure of intimate partner violence and suicidal ideation led to the inclusion of referral information for national services addressing gender-based violence and suicide prevention in the manual. Another case involving unplanned pregnancy highlighted the need for a structured closing session to support emotional closure and future planning.

Feedback from a consultative workshop informed further adaptations, including locally resonant examples for cognitive restructuring, emphasis on skill-based interactive training and suggestions to form Community Advisory Boards (CABs) to strengthen community participation and foster a sense of shared ownership in the program. While the example-based adaptations were implemented, establishing a formal CAB was not feasible due to operational cost constraints.

A qualitative study conducted at the end of the pilot identified four key themes related to feasibility and acceptability of (1) FCHVs as delivery agents, (2) adequacy of session frequency, (3) THP content and (4) CBT techniques. FCHVs were widely accepted due to their familiarity within the community, and their visits were seen as part of routine care. While eight visits were generally acceptable, some participants expressed interest in additional antenatal sessions. Participants found the THP content and the visual

aids relevant, although some FCHVs recommended simplifying language. Exercises like walking and deep breathing were particularly valued by women. CBT technique was initially difficult to understand but with practice, FCHVs were able to understand. Findings from the competency assessment are reported elsewhere (manuscript forthcoming). Further illustrative quotes are provided in Table 3.

Several substantive adaptations made after the program implementation include integrating narrative examples, adding discussion questions (e.g., "How can the family help the woman during her pregnancy/postnatal period?") as opposed to step-by-step instructions and generic prompts such as "allow time for discussion." Additional health information on constituents of balanced diet using examples of locally available food sources, information on the benefits of exercise listed in the health calendar, and importance of regular health checkup were added which were missing in the original THP manual. An overview of key tasks and learning objectives, which also serve as a checklist, was added on the first page of each session. Finally, visual aids were redesigned by aligning images horizontally rather than vertically, thereby improving clarity and illustrating connections between concepts more effectively.

## Discussion

This study responds to recommendations from previous research, which highlighted the need for participatory and multidisciplinary approaches when adapting psychological interventions for perinatal depression in Nepal (Subba et al., 2024). Following the mhCACI framework and adhering to eight domains of the EVM (Bernal et al., 1995; Bernal and Sáez-Santiago, 2006), we tailored the

content, delivery mechanisms and implementation strategies. FCHVs were chosen for THP delivery. Key adaptations included condensing the original five modules into two modules to fit the FCHVs' routine schedule, adding supplementary content on communication skills, incorporating a closing session and replacing technical/clinical language with culturally resonant idioms and simplified descriptions. Additional changes included embedding local examples, referral pathways and discussion prompts, as well as modifying the manual's layout such as adding visual illustrations and a summary table of tasks per session to improve usability and contextual relevance. Further, the adapted intervention was also qualitatively assessed for feasibility and acceptability with both community and health system actors.

We chose FCHVs for THP delivery for several reasons. First, as members of Nepal's public health system (Panday et al., 2017; Ministry of Health and Population Nepal, 2019), FCHVs eliminate the need to introduce a new workforce, aligning with WHO's recommendation to integrate perinatal mental health into existing MCH services (WHO, 2022). Embedding THP within routine programs also supports long-term sustainability (Subba et al., 2025). Second, FCHVs maintain regular contact with pregnant and postnatal women and are widely trusted within their communities. Our qualitative findings showed that their visits were perceived as part of routine care, reducing stigma often associated with mental health interventions. In tight-knit communities, such integration helped normalize THP visits. Third, FCHVs possess a strong contextual and cultural knowledge of their communities. Their shared characteristics with service users contributed to increased comfort and acceptability. This relational similarity has been shown to foster trust and enhance the effectiveness of community-based care in both high- and low-income settings (Subba et al., 2025).

We reduced the number of THP sessions from 16 to 8 to align with the FCHVs' routine schedule of four antenatal and four postnatal visits. Similar adaptations have been made in countries like Vietnam (Fisher et al., 2014), Malawi (Ng'oma et al., 2024), China (Nisar et al., 2020), Peru (Aguilar et al., 2023), Turkey (Boran et al., 2023a) and Pakistan (Sikander et al., 2019), where the number of sessions was adjusted to suit local contexts. A recent trial that condensed THP to 8 sessions found it to be just as effective in reducing depressive symptoms and improving maternal health outcomes (Rahman et al., 2025). In low-resource settings, where long interventions and limited provider capacity can limit scalability, this adaptation offers a more feasible and sustainable delivery model.

Another significant adaptation was the introduction of FHS training. While the original THP manual lists communication skills as a prerequisite for delivery agents, it does not provide guidance on what these skills entail. In consultation with the intervention developers, we added a 2.5 days (approximately 15 h) training module on FHS prior to THP training. The FHS curriculum (Watts et al., 2021; Pedersen et al., 2023), although designed for 20 h, allowed flexibility to tailor the content to our context. Our key focus during the FHS training was on building skills in attitude of helpers, nonverbal and verbal communication, confidentiality, empathy and fostering hope for change which are essential for quality care in task-sharing models. Based on our experience, we recommend that future adaptations of THP or similar interventions incorporate FHS as a foundational training component.

Substantial adaptations were made to improve the relevance of the content and reduce stigma. For example, a subsection on stigma, misconceptions and harmful beliefs surrounding perinatal depression was added to the background section, based on the findings from a previous qualitative study in Nepal (Subba et al., 2024). Consistent with previous studies, we used commonly used terms in Nepali communities such as *maan ko samasya* (heart-mind problem) and tension to describe emotional distress, which are culturally more appropriate and acceptable (Kohrt and Harper, 2008; Jordans et al., 2015; Subba et al., 2017; Sangraula et al., 2021).

This study encountered several challenges during the adaptation process. While we made important adaptations to increase the cultural and contextual relevance of the THP, we were also unable to implement every recommendation. For example, although forming CAB was deemed essential, it was not feasible due to resource limitations, instead, we engaged with diverse local stakeholders through informal discussions, roundtables and field visits with FCHVs, community members and health officials throughout the intervention implementation. Feedback from the feasibility study such as the need for more antenatal sessions was not addressed primarily because FCHV is a voluntary position and expanding their workload without remuneration posed ethical concerns for the study team. Recent studies globally have also raised critiques about the unpaid nature of community health workers, emphasizing the need for fair compensation (Chase et al., 2022; Smithwick et al., 2023; Tikkanen et al., 2024). Arranging booster sessions or targeted follow-ups for women can be some potential options.

Similarly, the suggested change in terminology from "positive" and "negative" to "good" and "bad" thoughts was not incorporated. While some FCHVs found the latter simpler, these terms risk moral or subjective interpretation. Therefore, "positive" and "negative" thoughts was retained to reflect the impact of thoughts, whether they lead to constructive or distressing outcomes rather than moral judgment. FCHVs also faced initial challenges in applying the CBT model, as perinatal distress in the Nepali context is typically expressed through emotions or somatic symptoms than thoughts (Subba et al., 2024). Linking emotions to triggering events helped FCHVs identify underlying negative thoughts more effectively. This adaptation was applied during the implementation phase.

These challenges highlight a central dilemma in cultural adaptation process on determining *"what to change"* to enhance local relevance and *"what to retain"* to preserve the intervention's theoretical integrity (Castro et al., 2004; Heim and Kohrt, 2019). To navigate these decisions, adaptations were made cautiously and in consultation with the THP developers (co-authors NA, AR, SS).

THP's core ingredients have been broadly categorized into its psychological components (e.g., CBT strategies, problem-solving) and social components (e.g., empathic listening, strengthening family support) (Rahman et al., 2025). Mediation analyses from randomized controlled trials in India and Pakistan have shown behavioral activation and social support to be key mechanisms of change in both contexts (Singla et al., 2021). In Nepal, however, we only evaluated competency of the FCHVs (manuscript forthcoming). While essential, competency assessment alone does not capture potential mediators. As a result, it remains unclear which specific therapeutic ingredients drive improvements in our context. Future research can therefore incorporate mediator and moderator analyses to disentangle how different psychological and social components contribute to change, and to identify which processes matter most across different cultural settings. Such evidence will be critical for refining THP's theory of change and informing future adaptation and scale-up.

The mhCACI framework enabled us to embed local culture and context into both the intervention content and implementation strategy, enhancing engagement and acceptability among service

providers and users. While the framework's systematic approach ensured cultural relevance and appropriateness, its application proved time- and resource-intensive. Further, the steps could not be followed in a strictly linear sequence in practice. For example, the practice round began shortly after the training-of-trainers and the desk review and expert consultation spanned across both Phases 2 and 3. Our experience suggests that the mhCACI framework is best understood as a flexible framework to inform and shape the adaptation process rather than a rigid, step-by-step guide.

## Conclusion

We present the first adaptation of the WHO THP intervention for perinatal depression in Nepal. The intervention was rigorously adapted to align with the current Nepali health system. Our study demonstrated that the adapted THP is feasible and acceptable, but we recognize adaptation as an evolving process, not a fixed endpoint. Therefore, while this version of THP has been adapted for the Nepali context, considering the diverse sociopolitical, cultural, and religious practices in Nepal, minor adjustments may be necessary to accommodate the beliefs of the target population. Further, a definitive trial, including a cost analysis component, is needed to assess the effectiveness and cost-effectiveness of THP before its formal integration into the healthcare system.

### Abbreviations

| | |
|---|---|
| CAB | community advisory board |
| CBT | cognitive behavioral therapy |
| EVM | ecological validity model |
| FCHV | female community health volunteer |
| FHS | foundational helping skills |
| mhCACI | mental health cultural adaptation and contextualization for implementation |
| THP | thinking healthy programme |
| WHO | World Health Organization |

**Open peer review.** To view the open peer review materials for this article, please visit http://doi.org/10.1017/gmh.2025.10127.

**Supplementary material.** The supplementary material for this article can be found at http://doi.org/10.1017/gmh.2025.10127.

**Data availability statement.** Data used and/or analyzed during the current study are available from the corresponding author upon reasonable request.

**Acknowledgments.** We want to thank all the study participants and stakeholders who have supported during the intervention adaptation process. We also thank the ENHANCE Collaborative Learning Group members for their continuous support and feedback during the study process.

**Author contribution.** Prasansa Subba: Writing–original draft, Investigation, Formal analysis, Data curation, Conceptualization, Project Administration, Visualization; Pragya Shrestha: Conceptualization, Methodology, Investigation; Rupa Rai: Investigation, Project Administration; Shristi Subedi: Investigation, Writing – review & editing, Project Administration; Nagendra Prasad Luitel: Conceptualization, Supervision, Writing – review & editing; Atif Rahman: Supervision, Funding acquisition, Writing – review & editing; Siham Sikander: Supervision, Writing – review & editing; Najia Atif: Supervision, Conceptualization, Writing – review and editing.

**Financial support.** This study is supported under financial aid from the National Institute for Health and Care Research (NIHR), UK's RIGHT CALL 2 NIHR200817 ENHANCE: Scaling-up Care for Perinatal Depression through Technological Enhancements to the 'Thinking Healthy Program' (RIGHT CALL 2 NIHR200817). Further information available at: https://fundingawards. nihr.ac.uk/award/NIHR200817. The funding agency has no role in the collection, management, analysis and interpretation of data or the decision to submit the report of publication.

**Competing interests.** The authors declare that they have no known competing financial interests or personal relationships that could have appeared to influence the work reported in this article.

**Ethics statement.** The study was approved by the Ethical Review Board of Nepal Health Research Council (Ref: 276/2021). A separate ethical application was submitted and approved by the Ethical Review Board of Nepal Health Research Council (Ref: 4227) and the Central University Research Ethics Committee (Ref: 11150) for the pilot testing of the adapted THP. Study procedures followed were in accordance with the ethical standards of the responsible committee and with the Helsinki Declaration of 1975 and its later amendments. Written informed consent was obtained from all individual study participants prior to the interviews.

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
