## [Reviewer Report]

The article is very well written. A lot of work had to be synthesized within an allocated word limit and this was well done. A few suggestions for improving the manuscript:

The authors use the mhCACI framework which supports rigour in approach. However, the order in which mhCACI suggests to follow and that of which happens in a timely/logical real-world order makes it confusing to comprehend in this manuscript exactly how steps influence one another. For instance

a. In step 3, ToT, it writes that 2 people are trained and then train the FHCV, but afterwards, the manual is translated. This raises the question: Why wasn’t the manual translated prior to the ToT? Also, do we need to write about the training of the FCHvs in that section? It is confusing whether, in fact, it first went through with feedback from FCHV before manual translation.

b. step 2: in depth literature review - what were the results and how were they applied for informing steps 2,4, 6, 7, 8 and 9?

c. step 4 - translation - was there a charting specifically of concepts that were documented and changed by the same translator or first mapped out in the ToT / Step 3?

d. step 6 - what is the approach to the analysis for this qual? Also, formative implies prior to training, but so far we have read about a ToT and cascaded model with FHCV adaptations (e.g., make content suited to FCHV education as written in Step 5), so was Step 6 done prior to or after the training of FCHVs?

e. Step 7 - practice rounds are following the ToT, yes? If so, I wonder why step 7 is placed under implementation when it is still a practice / pilot for the translated manual?

f. It was nice to have the baptism workshop, but how was it organized re: content to focus on? Was there a chart at this stage of what content you wanted them to review and give feedback on? What happened in the workshop that would be different from a reading of the manual, and how were those pieces selected for this workshop? Also, here, participants omit FCHV, so was this after qual feedback with FCHV but prior to training them?

g. here FHS training is mentioned but there is no reference. It would be ideal to reference the article co-authored by one of the authors of this current submission, which shows the iterative development of FHS in the Nepali context and its evidence for success (Pedersen, Shrestha et al 2023), and cite this as a key component for already having been reflective of the Nepali context. Also in this section, it is written that there is a qualitative study embedded in a pilot cluster randomized controlled trial. Which pilot RCT? What stage is that in the framework? If there is an RCT, giving context as to why it is mentioned here would be useful. Also, the only analysis offered for the qual is a framework approach, but more information on how the analysis was done would be very useful to support the results.

h. final refinements - made after which qual study, the one in the formative phase or the one mentioned as part of an RCT?

Other suggestions: the authors use initials, but when first mentioning them please write “authors”; and in the case of first mentioning PS1 and PS2, perhaps refer to them as trainees, explicitly saying that there was a ToT “of 2 trainees, authors PS1 and PS2.” It’s important to note a # of trainees, for context of readers and feasibility.

In 3.3 Phase 3 implementation, it mentions the CAB suggestion. But doesn’t specify whether this suggestion was intended to implement a CAB for sustained THP, or simply for a study?

Discussion: It would be great to discuss real-world implications, specifically the applicability/feasibility of applying mhCACI framework. The authors make some mention, but only to their adaptations in the process rather than reflecting on real world practice more broadly. Similarly, I see the comment on how changing terminology from positive to negative to good and bad was not considered as preserving the theoretical foundation, but I was under the impression it was “helpful and unhelpful” which preserves the theory and moves away from a potentially stigmatising framing of thoughts and feelings, such as can be the framing “positive and negative” or “good and bad.” Would authors clarify this? Similarly, mention towards how mechanisms of change are being measured or tested e.g., competency+fidelity with client outcomes & feedback)

in future research would be useful in terms of testing the original THP theorectical model / mechanisms for change with new adaptations, and how this aligns with the decision-making process for adherence to these models through the mhCACI process.

Finally, apologies for harping on the references, but again, there is a Watts reference for the FHS study, and I feel it would be more complete to add the Pedersen Shrestha 2023 paper, since it documents the Nepal iterative development, translation/contextualisation, and successful testing of the content and how it came to be 2.5 days, alongside two other. This could also help justify why the FHS did not need to undergo rigorous adaptation processes like THP.

conclusion - authors mention a definitive trial, but what about the pilot RCT? Since this paper does not present quant results of the RCT but mention it, I would suggest choosing whether the pilot RCT is mentioned prior to the conclusion, and if so, to mention that those results need to be analyzed and presented (quantitatively with the qual) prior to a definitive trial being suggested with CEA. Cost analysis could also be part of a pilot RCT

---

## [Reviewer Report]

This was an interesting and well thought through cultural adaptation of the THP that originally took place in Pakistan and was published in 2008. The following are some comments I hope will improve the quality of the manuscript.

1. In the introduction, first paragraph, authors describe that the prevalence for perinatal depression has more than doubled since 2017. I would suggest that authors closely examine both reviews including the selection criteria and included trials to understand why these estimates are so different. It may be one review is much high quality than the other, or different selection criteria were used.

2. Authors state that mechanisms of action were identified. I noticed a few things about this. Firstly, social support was not identified in potential mechanisms of action. This mechanism has been identified elsewhere as a key pathway in two mediation analyses. I would suggest that authors examine the potential for this pathway. Targeting levels of social support may help to improve recover from depression/reduce symptoms of perinatal depression. It also seemed a bit odd that people with lived experience were not involved with this. Involving people who have experienced of perinatal depression may help to better understand pathways through which an intervention can improve symptoms of depression.

3. Authors state that they did a systematic review of psychosocial interventions. I would suggest that they include psychological interventions in this review as well. Sometimes people who develop/test these interventions may define psychosocial interventions as psychological interventions.

4. Please spell out the acronym PI and PII - it was not possible to understand what authors meant as this was not spelled out in full.

5. For the formative qualitative study, I was wondering if people with lived experience were involved? It would be useful to understand women living or have lived with perinatal depression experiences in perinatal depression, etc.

---

## [Reviewer Report]

This is a clearly written article describing rigorous research on a psychological intervention for maternal mental health. It captures best practice in the field of global mental health, documenting a cultural adaptation process developed by a Nepali NGO for precisely this purpose. The research methods are well-described and sound. The findings will be of use to many researchers faced with similar challenges of cross-cultural difference in delivery of community mental health support. Overall, I strongly recommend publication of this article. However, there are two areas I would invite the authors to expand their analysis if they see fit, hence a decision of minor revisions:

First, there are areas where I felt some further explanation would be very helpful to the journal’s readership. For example, it is mentioned that the phrase “healthy thought” (which is in the title of the intervention and therefore presumably core to the theory involved) was changed to “positive thought”, but the reader is left wondering why this exactly this change was made and if any possible alteration of meaning was considered. Also, why exactly was it deemed inappropriate to further adapt this to “good thought” following participants’ input? What aspect of the intervention did “good” compromise that “positive” did not? Elucidating some of the choices made further (including reasoning and trade-offs considered) could greatly increase the value of the article to readers making similar kinds of adaptations in mental health practice.

Second, there seems a missed opportunity here to reflect more generally on issues of cross-cultural validity that have been at the heart of debates on global mental health. At the least, I would’ve loved to see a little bit more critical reflection on the limitations or compromises involved in cultural adaptation of a CBT model in Nepal, if any. The impression given is that the culturally adapted model of THP developed is a perfect/seamless fit for Nepal’s sociocultural context; do the authors feel this to be the case? I would be very curious to know if you identified any concepts or practices that could not be perfectly translated into Nepali, leading to compromises on content or meaning to make the intervention locally relevant. For example, you mention that people initially struggled to understand CBT content – what cultural understandings posed a challenge to this? Was the premise of CBT that people should challenge their negative thoughts uncritically accepted by participants? If this critical reflection is planned for another forum, it would be helpful to signal that or refer to forthcoming publications etc.

Thank you for the opportunity to review this interesting work, and I look forward to seeing it published!

---

## [Reviewer Report]

Many thanks for sharing this revised version of the manuscript. It was helpful to see some added reflection on decisions around how to label thoughts within the intervention. Whilst I would’ve been very interested to hear more of the authors' reflections on the limits of cultural adaptation in this context, I appreciate that this may not be the appropriate forum. I look forward to reading more work by this team in the future.

---

## [Reviewer Report]

Thank you for submitting the revised manuscript. The authors have been very responsive to previous feedback, and the manuscript is much improved. Congratulations again on this important work.

Regarding the rationale and significance laid out by the authors for including the FHS training prior to THP, I recommend reframing the first row of the mechanisms of action table from “empathic listening” to “foundational helping skills” or “common factors.” As it reads now, “empathic listening” suggests two skills, whereas its definition “foundational counselling” points to a broader set of 15 concrete skills of which are covered in the referred to FHS training. This framing would also be in line with the latest framing and research on mechanisms of change, particularly the distinction and harmony between “common” and “specific” therapeutic factors.

Alternatively, as previously suggested, the authors could address this in the discussion. They could clarify how mechanisms of action are measured in the current study with the added FHS training, compared to the individual THP training and analysis in Rahman 2025. Given the cited Rahman 2025 identifies perceived ‘active ingredients’ but does not analyse mediators or moderators, this manuscript has an opportunity to advocate for more comprehensive measurement of therapuetic mechanisms and their ingredients (e.g., continued feedback from mothers‘/participants’, providers fidelity to technique like social support or behavioral activation as laid out in THP, mothers‘ attendance to session, and providers’ behavioral performance of competency in both common and specific factors).

---

## [Editor Report]

Please would you kindly attend to the request by reviewer two for more detail with respect to a suggestion that they had made in the first review.